# Balenine, Imidazole Dipeptide Promotes Skeletal Muscle Regeneration by Regulating Phagocytosis Properties of Immune Cells

**DOI:** 10.3390/md20050313

**Published:** 2022-05-05

**Authors:** Min Yang, Luchuanyang Sun, Yasunosuke Kawabata, Fumihito Murayama, Takahiro Maegawa, Takeshi Nikawa, Katsuya Hirasaka

**Affiliations:** 1Graduate School of Fisheries and Environmental Sciences, Nagasaki University, Nagasaki 8528521, Japan; 18342221707@163.com (M.Y.); slcywelcome@126.com (L.S.); 2Kyokuyo Co., Ltd., Siogama 9850001, Japan; yasunosuke_kawabata@kyokuyo.co.jp (Y.K.); fumihito_murayama@kyokuyo.co.jp (F.M.); takahiro_maegawa@kyokuyo.co.jp (T.M.); 3Department of Nutritional Physiology, Institute of Medical Nutrition, Tokushima University Medical School, Tokushima 7708503, Japan; nikawa@tokushima-u.ac.jp; 4Organization for Marine Science and Technology, Nagasaki University, Nagasaki 8528521, Japan

**Keywords:** balenine, muscle regeneration, dipeptide, inflammation, phagocytic activity

## Abstract

Balenine is one of the endogenous imidazole dipeptides derived from marine products. It is composed of beta-alanine and 3-methyl-L-histidine, which exist mainly in the muscles of marine organisms. The physiological functions of dietary balenine are not well-known. In this study, we investigated whether the supplementation of dietary balenine was associated with muscle function in a cardiotoxin-indued muscle degeneration/regeneration model. Through morphological observation, we found that the supplementation of balenine-enriched extract promoted the regeneration stage. In addition, the expression of regeneration-related myogenic marker genes, such as paired box protein 7, MyoD1, myogenin, and Myh3, in a group of mice fed a balenine-enriched extract diet was higher than that in a group fed a normal diet. Moreover, the supplementation of balenine-enriched extract promoted the expression of anti-inflammatory cytokines as well as pro-inflammatory cytokines at the degeneration stage. Interestingly, phagocytic activity in the balenine group was significantly higher than that in the control group in vitro. These results suggest that balenine may promote the progress of muscle regeneration by increasing the phagocytic activity of macrophages.

## 1. Introduction

In vertebrates, there are three forms of imidazole dipeptides that are most commonly known: carnosine (β-alanyl-L-histidine), anserine (β-alanyl-3-methyl-L-histidine) and balenine (also called ophidine; β-alanyl-1-methyl-L-histidine), and these are found in skeletal muscle and the brain [1,2,3]. Among the imidazole dipeptides, balenine, a marine imidazole dipeptide exists, mainly in marine animals, opah (*Lampris guttatus*) and whales [4,5,6]. The chemical structure of balenine is shown in Figure 1. We previously reported that balenine could activate anti-oxidant enzymes in C2C12 myotubes and skeletal muscle [7]. Consistent with our results, Ishihara et al. reported that opah-derived balenine had a high anti-oxidant capacity and Fe (II) ion-chelating ability [8]. Thus, the supplementation of balenine had some effect on muscle function. In contrast to balenine, carnosine and anserine are known to exhibit various biological activities, such as an anti-oxide activity [9], the scavenging of free radicals [10], pH buffering [11], and sciatic nerve recovery effects [12]. The bioactivity and physiological function of balenine are largely unknown.

Skeletal muscle is the largest cellular compartment of the body, and an immunologically unique tissue. Muscle regeneration, which is induced by acute injury or chronic muscle disease, is accompanied by myofibril degradation and inflammation in skeletal muscle, resulting in repair and regeneration through the activation of satellite cells [13,14,15]. Inflammation is a complex and essential biological process in the body, especially for the repair of tissue damage; this process features pro-inflammation and anti-inflammation. Pro-inflammation in skeletal muscle damage involves the removal of necrotic cell and muscle debris, and the activation and proliferation of satellite cells, while anti-inflammation regulates muscle cell differentiation, fusing and forming muscle fibers [16]. Neutrophils and macrophages, which are the body’s first line of defense against invading microorganisms, first immigrate into peripheral tissues to eliminate dead cells and muscle debris [17]. After muscle injury, neutrophils speedily infiltrate impaired areas. Subsequently, pro-inflammatory macrophages (M1), which secrete cytokines, such as tumor necrosis factor (TNF)-α, interleukin (IL)-1β, inducible nitric oxide synthase, and the monocyte chemoattractant protein (MCP)-1, accelerate the activation and proliferation of satellite cells, and act on damaged skeletal muscle [18]. Conversely, IL-4, IL-10, IL-13, and transforming growth factor (TGF)-β1 are secreted by anti-inflammatory macrophages (M2), leading to the acceleration of myoblast differentiation.

It was recently reported that the treatment of carnosine can promote the phagocytic capacity of macrophages to eliminate senescent skin cells [19]. Given the importance of phagocytic modulation by active compounds such as peptides, balenine may also regulate immune responses to clear degenerated organs. In this study, we examined the effect of the marine imidazole dipeptide and balenine on immune capacity through a muscle degeneration/regeneration model.

## 2. Results

### 2.1. Effect of Dietary Balenine-Enriched Extract on Wet Weight and Morphological Change in Muscle in a CTX-Indued Muscle Degeneration/Regeneration Model

It has been reported that Cardiotoxin (CTX) induces acute muscle injury with the loss of muscle weight and degradation of muscle fibers [20]. Consistent with that report, the wet weight of CTX-injected tibialis anterior (TA) muscle in the normal diet group was significantly decreased at day 3, compared with day 0 (Figure 2; degeneration stage), but subsequently increased at days 7 and 14 (Figure 2; regeneration stage). The wet weight of CTX-injected TA muscle in the balenine-enriched extract diet group at day 14 was significantly increased, compared to day 0, but there were no significant differences among days 0, 3, and 7. At day 3, the wet weight of CTX-injected TA muscle in the balenine-enriched extract diet group was significantly higher than that in the normal diet group (Figure 2).

We next investigated the myofiber size of TA muscle by immunofluorescence staining for laminin, which is localized at the myofiber basal lamina, and for Hoechst33342 that stains nuclei. The cross-sectional area (CSA) of myofibers surrounding laminin-positive immunostaining in the non-CTX-injected muscle of the balenine-enriched extract diet group was similar to that observed in the normal diet group (Figure 3a; day 0). On day 3, after CTX injection into the muscle, Hoechst33342-positive cells in muscle fibers were observed in both diet groups, indicating that immune cells, such as neutrophils, macrophages, and T cells, had infiltrated the muscle to eliminate muscle debris (Figure 3a,b; day 3). Indeed, the number of Hoechst33342-positive nuclei on day 3 after CX injection was significantly increased, compared with the non-CTX injected group (Figure 3b). On days 7 and 14 after CTX injection into the muscle, in both groups, the myofibers contained central nuclei, which suggests that they were regenerating myofibers (Figure 3a,b; days 7, 14). Although morphological changes in TA muscle in the normal diet and balenine-enriched extract diet groups did not appear at days 0 and 14 after CTX injection, the alteration of infiltrated immune cells and the capacity of muscle regeneration by administration of balenine-enriched extract were observed at days 3 and 7, respectively.

### 2.2. Effect of Dietary Balenine-Enriched Extract on Regeneration Stage in CTX-Indued Muscle

We examined the effect of balenine on the muscle regeneration stage. Interestingly, the cell-to-cell membrane space in the balenine-enriched extract diet group was narrower than that in the normal diet group on day 7 after CTX injection (Figure 4a). The average cell-to-cell membrane space of the balenine-enriched extract diet group was approximately 40% of that of the normal diet group (Figure 4a). In contrast, there was no significant difference between the two groups in the CSA of the myofibers contained in the central nuclei.

To confirm the morphological phenotype in the administration of balenine-enriched extract, we measured the expression of muscle regeneration marker genes. The expressions of paired box protein (Pax7; a transcription factor for myogenesis to regulate proliferation of precursor cells), mRNA in muscle in the normal diet and balenine-enriched extract diet groups reached their peak values on day 3 after CTX injection. These expressions subsequently decreased on days 7 and 14. The expression of Pax7 mRNA in the balenine-enriched extract diet group was significantly higher than that in the normal diet group (Figure 4b). Consistent with the expression pattern of Pax7, the expressions of MyoD1 and myogenin (which activate their transcription during muscle differentiation), mRNA in muscle in the normal diet and balenine-enriched extract diet groups were at their maximums on day 3 after CTX injection. The expressions of MyoD1 and myogenin mRNA in the balenine-enriched extract diet group were higher than those in the normal diet group (Figure 4b). On the other hand, the expression of Myh3, which is expressed in regenerated myofibers [21], mRNA in muscle in the normal diet and balenine-enriched extract diet groups was at the maximum on day 7 after CTX injection. The expression of Myh3 mRNA in the balenine-enriched extract diet group was higher than that in the normal diet group (Figure 4b). These results raise the possibility that the supplementation of balenine-enriched extract promotes muscle regeneration.
Figure 4Effect of dietary balenine-enriched extract on muscle regeneration in a CTX-indued muscle degeneration/regeneration model. (**a**) Sections (5 μm thick) of TA muscle from the normal diet and balenine-enriched extract diet groups were immunofluorescence stained with anti-laminin antibody and Hoechst33342. Red and blue indicate laminin and nuclei, respectively. Scale bar = 100 μm. Magnification is ×40. The cell-to-cell membrane space, which is the area remaining after removing the muscle fiber area, and the CSA were calculated by BZ-II analysis software and counted in 13 high-power fields in 3 individual TA muscle sections. Data are means ± SD. Statistical analysis consisted of a one-way ANOVA and Tukey’s test. ** *p* < 0.01, compared with the normal diet group, on day 7 after CTX injection. (**b**) Total RNAs from TA muscle were extracted and subjected to real-time reverse transcription-polymerase chain reactions. The ratio between the intensities of myogenetic genes and 18S ribosomal RNA was calculated. Data are means ± SD (*n* = 3). Statistical analysis consisted of a one-way ANOVA and Tukey’s test. * *p* < 0.05, compared with the normal diet group, on day 3 after CTX injection. ND, mice fed the normal diet; Bal, mice fed the balenine-enriched extract diet.
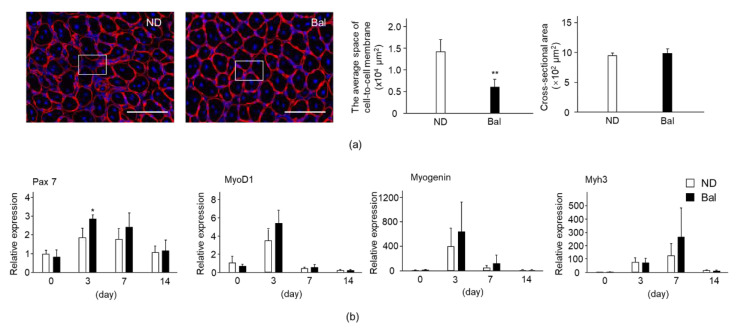


### 2.3. Effect of Dietary Balenine-Enriched Extract on Degeneration Stage in CTX-Indued Muscle

Immune cells, especially macrophages, which infiltrated into muscle at an early stage of muscle impairment, play an important role in muscle degeneration/regeneration [22]. To investigate the accumulation of immune cells into TA muscle, we examined immunofluorescence staining for M1 and M2 macrophage markers, such as CD86 and CD206, respectively. The immunoactivities of CD86 and CD206 in muscle in the normal diet and balenine-enriched extract diet groups were detected at day 3 after CTX injection (Figure 5a), whereas their immunoactivities were hardly detectable at day 0 (non-CTX injection), indicating that M1 and M2 macrophages migrating from the circulating system or residual cells into impaired muscle contributed to muscle repair.

Next, we measured the expression of cytokines and chemokine in CTX-injected muscle. The expressions of pro-inflammatory cytokines, such as TNF-α and IL-6, anti-inflammatory cytokine, TGF-β1, and MCP-1 mRNA in muscle in the normal diet and balenine-enriched extract diet groups reached their peak values on day 3 after CTX injection, and thereafter decreased on days 7 and 14 (Figure 5b). The expressions of TNF-α, MCP-1, and TGF-β1 mRNA in the balenine-enriched extract diet group were significantly higher than those in the normal diet group on day 3. The expression of anti-inflammatory cytokine IL-10 mRNA in the normal diet group gradually increased after CTX injection and reached its peak value on day 7. In contrast, the mRNA level of IL-10 in the balenine-enriched extract diet group reached its peak value on day 3, and was significantly higher than that in the normal diet group on that day (Figure 5b). These data indicate that balenine-enriched extract promotes the infiltration of pro- and anti-inflammatory-related immune cells into impaired muscle.
Figure 5Effect of dietary balenine-enriched extract on the infiltration of macrophages into muscle degeneration in a CTX-indued muscle degeneration/regeneration model. (**a**) Sections (5 μm thick) of TA muscle from the normal diet and balenine-enriched extract diet groups were immunofluorescence stained with anti-CD86 and CD206 antibodies, and stained with Hoechst33342. Green, red, and blue indicate CD86, CD206, and nuclei, respectively. Scale bar = 20 μm. Magnification is ×100. (**b**) Total RNAs from TA muscle were extracted and subjected to real-time reverse transcription-polymerase chain reactions. The ratio between the intensities of inflammatory factor related genes and 18S ribosomal RNA was calculated. Data are means ± SD (*n* = 3). Statistical analysis consisted of a one-way ANOVA and Tukey’s test. * *p* < 0.05, compared with the normal diet group, on day 3 after CTX injection. ND, mice fed the normal diet; Bal, mice fed the balenine-enriched extract diet.
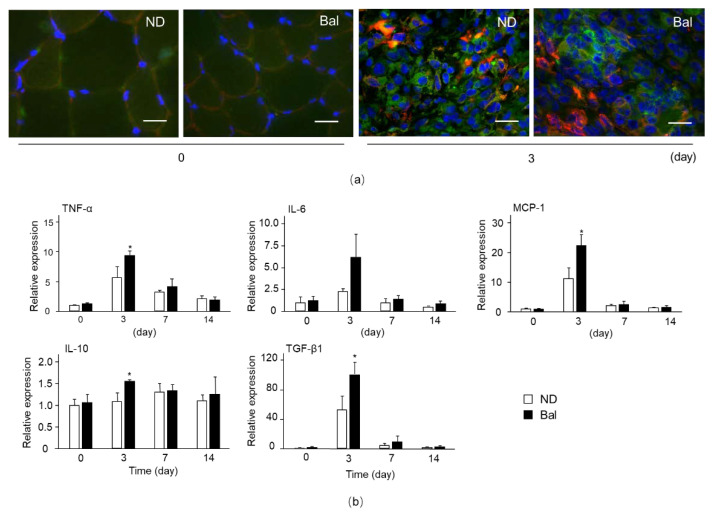


### 2.4. Effect of Balenine on Phagocytic Activity 

Balenine promoted the infiltration of both anti-inflammatory and pro-inflammatory-related immune cells into injured muscle. It was reported that the pro-inflammatory profile to anti-inflammatory profile of macrophages were associated with phagocytosis in the first step in the elimination of cell debris [23]. To confirm the effect of balenine on phagocytosis, we examined phagocytic activity using RAW264.7 cells. In neutral red staining, the phagocytic activity in RAW264.7 cells was gradually increased by balenine in a dose-dependent manner. In particular, the phagocytic activity of 10 mM balenine was significantly higher than that in the control group (Figure 6a). After incubation of opsonized *E. coli* BioParticales in RAW264.7 cells, particle uptake began in the cells. The phagocytosis properties of the 10 mM balenine-treated group were 2.38 times those of the control group (Figure 6b), suggesting that balenine treatment increased the phagocytosis properties of macrophages.

## 3. Discussion

In this study, we showed that the dietary intake of balenine-enriched extract inhibited the loss of muscle wet weight and promoted muscle regeneration in a muscle injury model. Muscle injury was accompanied by muscle weight loss and muscle fiber breakdown. Subsequently, muscle cell debris was removed by proteolysis [20,24]. Pax7, as a marker of satellite cell proliferation, was expressed at an early stage of muscle injury [25,26]. MyoD1 and myogenin, as myogenic regulatory factors, were expressed at the stage of differentiation [25,26]. Additionally, Myh3 (embryonic myosin heavy chain) was expressed in regenerated myofibers [21]. Balenine-enriched extract promoted gene expression at the stages of both proliferation and differentiation. Generally, the loss of muscle weight was found in muscle injury at an early stage [20,27,28]. In our results, for the balenine-enriched extract diet group, the wet weight of TA muscle did not decrease at day 3 after CTX injection (Figure 2), meaning that it might have had a beneficial effect at an early stage of muscle injury.

In muscle regeneration, inflammation factors, secreted by immune cells, are essential for the activation of satellite cells. Immune cells are carried to the site of muscle injury through blood circulation. Neutrophils and macrophages first reach the injury site post-injury, and remove dead cells and muscle debris [29]. Macrophages continue to eliminate apoptotic cells and muscle debris as phagocytes, while secreting inflammatory factors to activate satellite cells and regulate muscle regeneration [24]. Macrophages are mainly divided into pro-inflammatory M1 macrophages and anti-inflammatory M2 macrophages [30]. Although cytokines (TNF-α, IL-6) and chemokine (MCP-1), which are mainly secreted by M1 macrophages, are pro-inflammatory factors, they play important roles in the activation and proliferation of satellite cells [23,31]. Cytokines, such as IL-10 and TGF-β1, secreted by M2 macrophages, play a major role in the anti-inflammatory stage [32,33]. These cytokines and chemokine reduce local inflammation and contribute to angiogenesis, myoblast fusion, and new myofiber formation. We detected CD86- and CD206-positive cells at day 3 after CTX injection in the normal diet group and the balenine-enriched extract diet group, while these cells were hardly detectable in non-CTX injected muscle (Figure 5a). On the other hand, the expression of pro- and anti-inflammation factors in the balenine-enriched extract diet group were higher than those in the normal diet group (Figure 5b). These findings raise the possibility that balenine-enriched extract contributes to activation of M1 macrophages’ properties, thereby promoting muscle regeneration.

Muscle debris were eliminated by the phagocytosis of macrophages, resulting in those macrophages releasing pro-inflammatory cytokines. In addition, the phagocytosis of muscle debris induces a switch in macrophages, from a pro-inflammatory M1 macrophages profile to an anti-inflammatory M2 macrophages profile releasing TGF-β [23]. Indeed, the deficiency of transglutaminase 2, which is a versatile enzyme participating in efferocytosis, showed smaller sizes of regenerating fibers and delayed myoblast fusion after CTX injection in TA muscle [34]. Thus, the phagocytosis properties of macrophages played an important role in muscle repair after injury [35]. In this study, we demonstrated that the phagocytic activity of RAW264.7 cells was increased by balenine treatment. Consistent with our results, it was reported that carnosine, an imidazole peptide, could enhance the phagocytic activity of macrophages through an activation of the AKT2 signaling pathway by CD36, a membrane protein that facilitates fatty acid uptake, and RAGE, a receptor for advanced glycation end products, to eliminate senescent skin cells [19]. However, further studies are required to identify the mechanism of phagocytosis activity by balenine treatment.

Sigemura et al. demonstrated that balenine was detected in the plasma of mice at 1 h after oral administration of opah-derived balenine [36]. Notably, the concentration of balenine in plasma was higher than that of carnosine. Everaert et al. reported that the supplementation of carnosine resulted in an increase in carnosine content in muscle and increased fatigue resistance [37]. Thus, the finding that balenine was also detected in the blood after oral administration suggested that balenine may act in peripheral tissues such as muscle. To prove the function of balenine in vivo, however, further experiments on the half-life of balenine and the presence/absence of balenine into filter organs are required.

It has been reported that aging induces impaired muscle regeneration through a decrease in muscle satellite cells and age-related changes in the immune system [38,39]. Meanwhile, several nutrients preserve the capacity for skeletal muscle regeneration with aging [40]. In our results, the supplementation of balenine-enriched extract promoted muscle regeneration via a stimulation of immune system. Thus, some nutrients might represent a novel therapeutic strategy for muscle injury in the musculoskeletal locomotor system.

## 4. Materials and Methods

### 4.1. Chemicals and Muscle Injury Model

CTX was purchased from Latoxan (Rosans, France). An extraction of dietary crude balenine prepared from opah meat was provided by Adaptgen Corp. (Tajimi, Japan). Its nutritional composition comprised 84.0% protein, including 27.3% balenine, 0.1% lipid, 11.8% ash, and 4.1% fluid.

Male C57BL/6J mice (Japan CLEA, Tokyo, Japan) aged six weeks were kept in a room maintained at 24 ± 1 °C on a cycle of 12 h light and 12 h dark, with food (Oriental Yeast Company, Tokyo, Japan) and water available ad libitum. Briefly, after acclimatization for one week, the mice were divided into two groups. They were given a 1% *w*/*w* concentration of balenine-enriched extract diet (the concentration of balenine described in the previous study) or a normal diet for two weeks, from seven weeks of age. Shigemura et al. reported that the administration of 200 mg/kg opah-derived balenine did not affect endogenous amino acid concentrations, including histidine and alanine in the blood [36]. The nutritional composition of each diet was the same as in a previous mice experiment [41]. The mice were then subjected to a muscle degeneration/regeneration model. Briefly, CTX (10 μM in 100 μL of saline) was injected into both left and right TA muscle. During the development of muscle degeneration/regeneration, the mice continued to receive the normal diet or the balenine-enriched extract diet until the termination of the experiment, after 14 days. The TA was isolated at the time of sacrifice at 0, 3, 7, and 14 days, respectively. After the wet weight of TA was measured, it was immediately frozen in chilled isopentane and liquid nitrogen and stored at −80 °C until analysis. All animal experiments involving balenine-enriched extract supplement and muscle degeneration/regeneration were approved by the Committee on Animal Experiments of Nagasaki University and were performed in accordance with the guidelines for the care and use of laboratory animals prescribed by the university (Permission No. 1803291443-4).

### 4.2. Immunofluorescence Staining

The isolated TA muscles of mice were immediately frozen in chilled isopentane and liquid nitrogen and stored at −80 °C until analysis. The mid-belly transverse cryosections of TA muscles (5 μm thick) were prepared with a Leica CM1950 cryostat (Wetzlar, Germany). Sections were placed on slide glasses coated with poly-L-lysine, fixed in ice-cold acetone, and stained using an immunofluorescence antibody. The primary antibody reaction was performed using anti-laminin (abcam, ab11575, Cambridge, UK). The second antibody (Thermo Fisher Scientific, Waltham, MA, USA) reaction was performed using anti-mouse Alexa Fluor 568 IgG (red). The nucleus was stained with Hoechst33342 (blue). Images were acquired using a BIOREVO BZ-X710 fluorescence microscope (Keyence, Osaka, Japan) and a camera, and then processed using BZ-X Analyzer with BZ-H3A Advanced Application software (Keyence, Osaka, Japan).

### 4.3. Quantitative Reverse Transcription (RT)-Polymerase Chain Reaction (PCR)

Total RNA was extracted from mouse TA muscle using an acid guanidinium thiocyanate–phenol–chloroform mixture (ISOGEN™; Nippon Gene, Tokyo, Japan). Quantitative RT-PCR was performed with the appropriate primers and SYBR^®^ green dye, using a real-time PCR system (ABI Real-Time PCR Detection System; Applied Biosystems, Foster City, CA, USA). The oligonucleotide primers used for PCR are shown as follows. IL-6: S 5′-CCGGAGAGGAGACTTCACAG-3′, AS 5′-TCCACGATTTCCCAGAGAAC-3′; IL-10: S 5′-CAGAGAAGCATGGCCCAGAA-3′, AS 5′-GCTCCACTGCCTTGCTCTTA-3′; MCP-1: S 5′-GCCAGCTCTCTCTTCCTCCA-3′, AS 5′-GAGTAGCAGCAGGTGAGTGG-3′; Myh3 S 5′-CCAACAGACTCCTGGCACAT-3′, AS 5′-CTGAACAGTGCAGAGACGGT-3′, MyoD1 S 5′-GCTACGACACCGCCTACTAC-3′, AS 5′-GAGATGCGCTCCACTATGCT-3′; Myogenin S 5′-GAGGAAGTCTGTGTCGGTGG-3′, AS 5′-ATCTCCACTTTAGGCAGCCG-3′; Pax7 S 5′-CGCCATCAACCATGCATCAG-3′, AS 5′-TTCATGTGGTTGGAGGGAGC-3′; TGF-β1: S 5′-CTTTGTACAACAGCACCCGC-3′, AS 5′-CATAGATGGCGTTGTTGCGG-3′; TNF-α: S 5′-GGCCTCCCTCTCATCAGTTC-3′, AS 5′-CTTTGAGATCCATGCCGTTG-3′; and 18S: S 5′-GTAACCCGTTGAACCCCATT-3′, AS 5′-CCATCCAATCGGTAGTAGCG-3′. Finally, 18S ribosomal RNA was used as an internal standard gene.

### 4.4. Phagocytic Activity

The RAW264.7 macrophage cell line (ATCC, Rockville, MD, USA) was maintained in Dulbecco’s Modified Eagle’s Medium (DMEM) containing 10% fetal bovine serum (FBS), 100 U/mL penicillin, and 100 mg/mL streptomycin at 37 °C under a 5% carbon dioxide, 95% air mixture. For phagocytic activity analysis using neutral red, RAW264.7 cells were seeded in a 96-well plate at a density of 5 × 10^4^/well for 24 h. Three concentrations (1, 5, and 10 mM) of synthetic balenine (Hamari Chemicals, Ltd. Osaka, Japan) were treated with RAW264.7 cells for 24 h. Subsequently, 10 µg/mL lipopolysaccharide (LPS) was treated with cells for 3 h. After being washed twice with phosphate-buffered saline (PBS), cells were cultured with 0.075% neutral red solution for 1 h. The cells were then washed twice more with PBS and treated with cell lysis buffer (1M acetic acid: ethanol = 1:1) for 1 h at room temperature. The optical density value of each well was measured at 570 nm, using a micro-plate reader. Cell phagocytic activity (%) was calculated as follows: ODs/ODc × 100, where ODs and ODc are the OD values of balenine (sample) and control wells, respectively. 

For phagocytosis properties using *E. coli* BioParticales, RAW264.7 cells were seeded in an eight-well chamber at a density of 3 × 10^4^/well for 24 h. RAW264.7 cells were then treated with PBS or 10 mM balenine for 24 h. After opsonization by mixing with culture-grade water for 1 h, 20 µg/mL *E. coli* BioParticales conjugated with Alexa Fluor 594 were treated in cells for 1 h at 37 °C. The cells were fixed with 4% paraformaldehyde for 10 min at room temperature. The phagocytosis properties of macrophages were analyzed using a BIOREVO BZ-X710 fluorescence microscope and a camera, and processed using BZ-X Analyzer with BZ-H3A Advanced Application software (Keyence, Osaka, Japan).

### 4.5. Statistical Analysis

All data were analyzed by a one-way analysis of variance, using SPSS statistics software, followed by Tukey’s test for individual differences between groups, and were expressed as mean ± SD. *p*-values of less than 0.05 were considered to indicate a significant difference.

## 5. Conclusions

In summary, it was demonstrated that balenine was associated with the progression of muscle regeneration through activation of phagocytosis capacity. In particular, balenine-enriched extract enhanced the intensity of M1 and M2 macrophages and increased the gene expression of cytokines (TNF-α, IL-6, MCP-1, IL-10, and TGF-β1) in the pro-inflammation and anti-inflammation stages, as well as phagocytic activity of macrophages, leading to facilitated muscle regeneration. These findings raise the possibility that the supplementation of dietary balenine-enriched extract promotes muscle regeneration following acute and chronic injury, such as post-exercise and aging.

## Figures and Tables

**Figure 1 marinedrugs-20-00313-f001:**
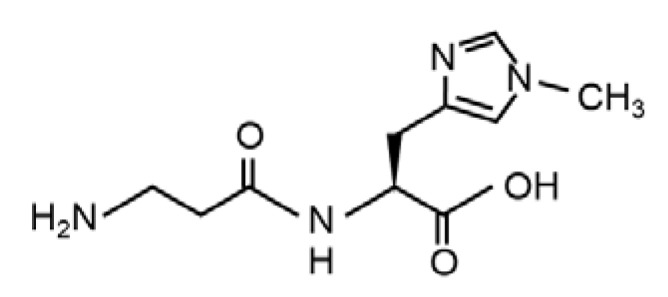
The chemical structure of balenine (β-alanyl-1-methyl-L-histidine).

**Figure 2 marinedrugs-20-00313-f002:**
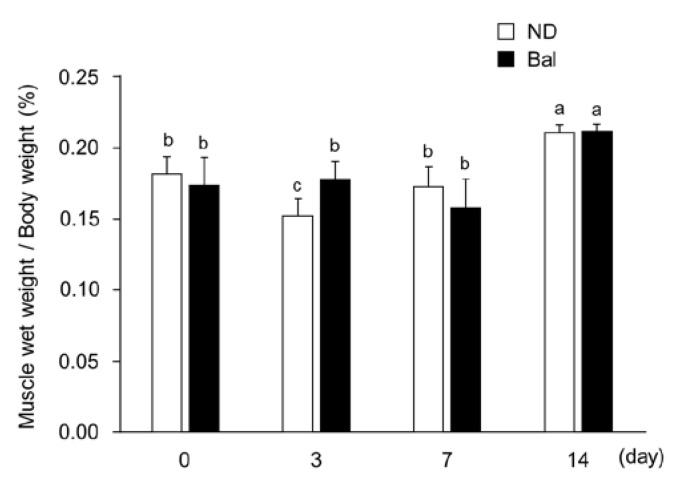
Effect of dietary balenine-enriched extract on wet weight of muscle in a CTX-indued muscle degeneration/regeneration model. A balenine-enriched extract-supplemented diet or normal diet was given to mice for two weeks. Their skeletal muscles were isolated at the indicated days after CTX injection. The wet weights of the TA muscle were measured. Data are presented as mean ± SD (*n* = 6). Different letters mean significant differences at *p* < 0.05. ND, mice fed the normal diet; Bal, mice fed the balenine-enriched extract diet.

**Figure 3 marinedrugs-20-00313-f003:**
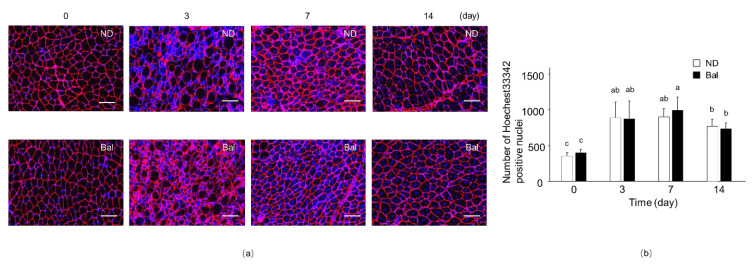
Time-dependent histological changes in skeletal muscle after CTX injection with balenine-enriched extract. (**a**) Sections (5 μm thick) of TA muscle from the normal diet and balenine-enriched extract diet groups were immunofluorescence stained with anti-laminin antibody, and stained with Hoechst33342. Red and blue indicate laminin and nuclei, respectively. Scale bar = 100 μm. Magnification is ×20. (**b**) The number of Hoechet33342-positive nuclei were counted in 13 high-power fields in three individual TA muscle sections and calculated by BZ-II analysis software. Data are means ± SD. Statistical analysis consisted of a one-way ANOVA and Tukey’s test. Different letters mean significant differences at *p* < 0.05. ND, mice fed the normal diet; Bal, mice fed the balenine-enriched extract diet.

**Figure 6 marinedrugs-20-00313-f006:**
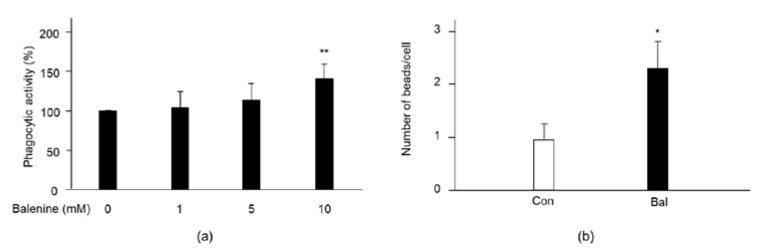
Effect of balenine on phagocytosis of RAW264.7 cells. (**a**) RAW264.7 cells were cultured with the indicated concentration of synthetic balenine for 24 h. Subsequently, 10 µg/mL lipopolysaccharide was treated with cells for 3 h. Cells were stained with 0.075% neutral red solution for 1 h, after which the optical density value of each well was measured at 570 nm, using a micro-plate reader. Cell phagocytic activity (%) was expressed by ODs/ODc × 100. ODs: OD values of the balenine wells. ODc: OD values of control wells. Data are means ± SD (*n* = 8). Statistical analysis consisted of a one-way ANOVA and Tukey’s test. ** *p* < 0.01, compared with the control group (0 mM balenine, PBS). (**b**) RAW264.7 cells were cultured with PBS or 10 mM balenine for 24 h. Opsonized 20 µg/mL *E. coli* BioParticales conjugated with Alexa Fluor 594 were treated in cells. One hour later, the number of beads in cells was counted in 12 high-power fields in 3 individual chambers, and expressed as means ± SD. * *p* < 0.05 compared with the control group (0 mM balenine, PBS). Con, control; Bal, synthetic balenine.

## Data Availability

Not applicable.

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
