# Peer review of "Balenine, Imidazole Dipeptide Promotes Skeletal Muscle Regeneration by Regulating Phagocytosis Properties of Immune Cells"

_marinedrugs, 2022, doi:10.3390/md20050313_

Round 1
Reviewer 1 Report
The authors investigated the facilitating effect of balnenin on muscle regeneration in a CTX-induced animal model. However, I have some questions prior to the publication of the study.
- Because balenin is a dipeptide molecule, it is highly possible to act as a nutrient source of amino acid. Therefore, it must be considered to eliminate the nutrient effect of balenin. For example, the control group should be treated with equivalent to the other dipeptide, which does not have myogenic bioactivity, Or, the control group must be treated with amino acids composed of alanin and histidine comparable amount of balenin in addition to the normal diet.
- The authors insisted that immune cells were infiltrated to the injured muscle lesion in Fig. 3 (line 101-102). It would be better to describe the method how you measured the infiltration of the immune cells by immunohistofluoresnce staining with laminin and to show the data in numeric (as a graph or table). If you performed H&E staining, it would be plausible to show the data.
- Page 5, line 165~167. and Page 6 Line 180~182. The authors claimed that Balenin diet induced the infiltration of immune cells into muscle lesion based on the Fig. 5. However, it is hard to conclude that balenin stimulated the infiltration of the immune cells. Because the authors did not investigate the source of the macrophages stained with CD86+ and CD206 from the circulating system or residual cells, in the study. Although the cytokines, TNF-a, TGF-b, and IL-10, are secreted from macrophages in the injured lesion in most cases, non-immune cells can be the sources of the inflammatory mediators. The authors must consider changing the description of the sentence.
It is hard to accept that Balenin changed the staining pattern of CD86 and CD206 in Fig. 5a. It would be better to show the fluorescence intensity of CD86 and CD206 in numeric. And it is necessary to show the pictures of fluorescence staining before and after CTX-injection.
- In Fig. 6a (line 195~196), did you treat the RAW264.7 cells with LPS or DMSO? Then which group did you treat with DMSO and LPS?
Author Response
Point-by-point response to the comments of Reviewer 1
We thank the reviewer for evaluating our manuscript. Thanks for the great suggestions and comments. Based on the comments, we provided more details and added Figures, result, discussion and materials and methods. The following text describes our response to the comments made by the reviewer. All line numbers mentioned in each response to each comment refer to the numbers that appear on the left margin of the text of the revised manuscript.
- Because balenin is a dipeptide molecule, it is highly possible to act as a nutrient source of amino acid. Therefore, it must be considered to eliminate the nutrient effect of balenin. For example, the control group should be treated with equivalent to the other dipeptide, which does not have myogenic bioactivity, Or, the control group must be treated with amino acids composed of alanin and histidine comparable amount of balenin in addition to the normal diet.
Response:
Thanks for your helpful comment. We agree with your comment. In this manuscript, we have used an extraction of dietary crude balenine prepared from opah meat for muscle de/regeneration model. Likewise, Shigemura et al., have been reported that the administration of 200 mg/kg opah-derived balenine did not affect endogenous amino acid concentrations including histidine and alanine in the blood (Shigemura et al., Foods 2022). In addition, the nutritional composition of crude balenine prepared from opah meat comprised 84.0% protein, including 27.3% balenine. Therefore, the crude balenine prepared from opah meat contains amino acids including histidine and alanine. Considering these information, we used casein as a control in this manuscript.
This information was added to the Material and Methods (Page 8, lines 290-292)
- The authors insisted that immune cells were infiltrated to the injured muscle lesion in Fig. 3 (line 101-102). It would be better to describe the method how you measured the infiltration of the immune cells by immunohistofluoresnce staining with laminin and to show the data in numeric (as a graph or table). If you performed H&E staining, it would be plausible to show the data.
Response:
Thanks for your helpful comment. To clear the number of infiltrated cells to the injured muscle lesion, we measured the number of Hoechet33342 positive nuclei by using BZ-II analysis software. The cross-sectional area (CSA) of myofibers surrounding laminin-positive im-munostaining in the non-CTX-injected muscle of the balenine-enriched extract diet group was similar to that observed in the normal diet group (Fig. 3a; day 0). On day 3 after CTX injection into the muscle, Hoechst33342 positive cells in muscle fibers were observed in both diet groups, indicating that the immune cells, such as neutrophils, macrophages, and T cells, had infiltrated muscle to eliminate muscle debris (Fig. 3a and b; day 3). Indeed, the number of Hoechst33342 positive nuclei on day 3 after CX injection was significantly increased, compared with the non-CTX injected group (Fig. 3b) On days 7 and 14 after CTX injection into the muscle, in both groups, the myofibers contained central nuclei, which suggests they were regenerating myofibers (Fig. 3a and b; days 7, 14).
These information are added to the Figure 3, result, discussion (Figure 3b, Page 3, lines 92-101, lines 111-114, Page 7, lines 242-245).
- Page 5, line 165~167. and Page 6 Line 180~182. The authors claimed that Balenin diet induced the infiltration of immune cells into muscle lesion based on the Fig. 5. However, it is hard to conclude that balenin stimulated the infiltration of the immune cells. Because the authors did not investigate the source of the macrophages stained with CD86+ and CD206 from the circulating system or residual cells, in the study. Although the cytokines, TNF-a, TGF-b, and IL-10, are secreted from macrophages in the injured lesion in most cases, non-immune cells can be the sources of the inflammatory mediators. The authors must consider changing the description of the sentence.
It is hard to accept that Balenin changed the staining pattern of CD86 and CD206 in Fig. 5a. It would be better to show the fluorescence intensity of CD86 and CD206 in numeric. And it is necessary to show the pictures of fluorescence staining before and after CTX-injection.
Response:
Thanks for your helpful comment. Based on your comment, we have improved the manuscript and added the pictures of fluorescence staining before and after CTX-injection in the Fig. 5a. The immunoactivities of CD86 and CD206 in muscle in the normal diet and balenine-enriched extract diet groups were detected at day 3 after CTX injection (Fig. 5a), whereas their immunoactivities were hardly detectable at day 0 (non-CTX injection), indicating that M1 and M2 macrophages migrating from the circulating system or residual cells into impaired muscle contribute to muscle repair.
These information are added to the Figure 5, result, figure legends (Figure 5a, Page 5, lines 161-165).
- In Fig. 6a (line 195~196), did you treat the RAW264.7 cells with LPS or DMSO? Then which group did you treat with DMSO and LPS?
Response:
Dear reviewer, we must apologize for this. This is a mistake. We have deleted DMSO. (Page 6, line 206).
Reviewer 2 Report
The paper entitled " Balenine, imidazole dipeptide promotes skeletal muscle regeneration by regulating phagocytosis properties of immune cells" explains how a diet based on balenine promotes muscle regeneration in a CTX murine model of muscle degeneration/regeneration.
Although the paper is well presented and discussed, it should be enriched by more accurate analyses also to elucidate the mechanisms with which the balenine can act.
Questions to be answered:
- what is balenine half-life?
- after how long is it internalized by the muscles and how much is directly eliminated in the kidney? it could be useful to analyse, also by qRT-PCR, the presence/absence of balenine into filter organs (i.e. liver, spleen, lungs)
- what are the pathways on which it could act?
- is there a difference between old mice and young mice?
- maybe it could be useful to add some Western Blots to evaluate more in details some markers of muscle regeneration, like MyoD or MyHCs.
For these reasons, major revision are required.
Author Response
Point-by-point response to the comments of Reviewer 2
We thank the reviewer for evaluating our manuscript. Thanks for the great suggestions and comments. Based on the comments, we provided more details and added Figures, result, discussion and materials and methods. The following text describes our response to the comments made by the reviewer. All line numbers mentioned in each response to each comment refer to the numbers that appear on the left margin of the text of the revised manuscript.
- what is balenine half-life?
- after how long is it internalized by the muscles and how much is directly eliminated in the kidney? it could be useful to analyse, also by qRT-PCR, the presence/absence of balenine into filter organs (i.e. liver, spleen, lungs)
Response:
Thanks for your helpful comment. Based on your comment, we have added the discussion about your suggestion. Sigemura et al. demonstrated that balenine was detected in the plasma of mice at 1-hour after oral administration of opah-derived balenine [Shigemura et al., Foods 2022]. Especially, the concentration of balenine in plasma was higher than that of carnosine. Everaert et al. reported that the supplementation of carnosine resulted in an increase of carnosine content in muscle and increased fatigue resistance [Everaert et al., Med Sci Sports Exerc. 2013]. Thus, the finding that balenine was also detected in the blood after oral administration suggests that balenine may act in peripheral tissues such as muscle. To prove the function of balenine in vivo, however, further experiments such as half-life of balenine and the presence/absence of balenine into filter organs are required.
This information was added to the discussion section (Page 7, lines 264-271).
- what are the pathways on which it could act?
Response:
Thanks for your helpful comment. We agree with your comment, since it is important information in this manuscript. It has been reported that carnosine, an imidazole peptide, could enhance the phagocytic activity of macrophages through an activation of AKT2 signaling pathway by CD36, a membrane protein that facilitates fatty acid uptake, and RAGE, a receptor for advanced glycation end products, to eliminate senescent skin cells (Li et al., Front. Pharmacol. 2020). Consistent with this finding, the phagocytic activity of RAW 264.7 cells was increased by balenine treatment. Therefore, balenine also may act via same pathway. However, further studies are required to identify the mechanism of phagocytosis ac-tivity by balenine treatment.
This information was added to the discussion section (Page 7, lines 258-263).
- is there a difference between old mice and young mice?
Response:
Based on your comment, we have added the discussion. It has been reported that aging induces impaired muscle regeneration through a decrease of muscle satellite cells and age-related changes in the immune system [Yamakawa et al., Int J Mol Sci. 2020, Tidball et al., Exp Gerontol. 2021]. Meanwhile, several nutrients preserve the capacity for skeletal muscle regeneration with aging [Domingues-Faria et al., Ageing Res Rev. 2016]. In our result, the supplementation of balenine-enriched extract pro-moted muscle regeneration via a stimulation of immune system. Thus, some nutrients might represent a novel therapeutic strategy for muscle injury in the musculoskeletal locomotor system.
This information was added to the discussion section (Pages 7-8, lines 272-278).
- maybe it could be useful to add some Western Blots to evaluate more in details some markers of muscle regeneration, like MyoD or MyHCs.
Response:
Thanks for your helpful comment. There were not enough samples to perform Western blots. There, we have added new data in the expression of embryonic MyHC. The expression of Myh3 (embryonic MyHC), which is expressed in regenerated myofibers [Zhang et al., Am J Pathol. 2009], mRNA in muscle in the normal diet and balenine-enriched extract diet groups was at the maxi-mum on day 7 after CTX injection. The expression of Myh3 mRNA in the bale-nine-enriched extract diet group was higher than that in the normal diet group (Fig. 4b).
These information are added to the Figure 4, abstract, result, discussion and materials and methods (Figure 4b, Page 1, line 23, Page 4, lines 136-140, Page 7, lines 224-225, Page 8, lines 324-325).
Round 2
Reviewer 1 Report
The authors replied sincerely against the questions raised by this reviewer. It seems that this manuscript is ready to be published.
Reviewer 2 Report
I would like to thanks the authors for the reply.
The paper is now ready for publication.